# Prevention of dementia using mobile phone applications (PRODEMOS): protocol for an international randomised controlled trial

Esmé Eggink ,[1] Melanie Hafdi,[2] Marieke P Hoevenaar-Blom,[2] Manshu Song,[3,4] Sandrine Andrieu,[5,6] Linda E Barnes,[7] Cindy Birck,[8] Rachael L Brooks,[7] Nicola Coley ,[5,6] Elizabeth Ford ,[9] Jean Georges,[8] Abraham van der Groep,[10] Willem A van Gool,[11] Ron Handels,[12,13] Haifeng Hou,[4,14] Dong Li,[14] Hongmei Liu,[15] Jihui Lyu ,[16] Harm van Marwijk ,[9] Mark van der Meijden,[10] Yixuan Niu,[17] Shanu Sadhwani,[9] Wenzhi Wang ,[15] Youxin Wang,[3,4] Anders Wimo,[13] Xiaoyan Ye,[18] Yueyi Yu,[19] Qiang Zeng,[20] Wei Zhang,[21] Wei Wang,[3,4] Carol Brayne,[7] Eric P Moll van Charante,[1,11] Edo Richard[11,22]

► Prepublication history and supplemental material for this paper is available online. To view these files, please visit the journal online (http://dx.doi.org/10.1136/bmjopen-2021-049762).

For numbered affiliations see end of article.

**Correspondence to**
Professor Edo Richard;
e.richard@amsterdamumc.nl

## ABSTRACT

**Introduction** Profiles of high risk for future dementia are well understood and are likely to concern mostly those in low-income and middle-income countries and people at greater disadvantage in high-income countries. Approximately 30%–40% of dementia cases have been estimated to be attributed to modifiable risk factors, including hypertension, smoking and sedentary lifestyle. Tailored interventions targeting these risk factors can potentially prevent or delay the onset of dementia. Mobile health (mHealth) improves accessibility of such prevention strategies in hard-to-reach populations while at the same time tailoring such approaches. In the current study, we will investigate the effectiveness and implementation of a coach-supported mHealth intervention, targeting dementia risk factors, to reduce dementia risk.

**Methods and analysis** The prevention of dementia using mobile phone applications (PRODEMOS) randomised controlled trial will follow an effectiveness–implementation hybrid design, taking place in the UK and China. People are eligible if they are 55–75 years old, of low socioeconomic status (UK) or from the general population (China); have ≥2 dementia risk factors; and own a smartphone. 2400 participants will be randomised to either a coach-supported, interactive mHealth platform, facilitating self-management of dementia risk factors, or a static control platform. The intervention and follow-up period will be 18 months. The primary effectiveness outcome is change in the previously validated Cardiovascular Risk Factors, Ageing and Incidence of Dementia dementia risk score. The main secondary outcomes include improvement of individual risk factors and cost-effectiveness. Implementation outcomes include acceptability, adoption, feasibility and sustainability of the intervention.

**Ethics and dissemination** The PRODEMOS trial is sponsored in the UK by the University of Cambridge and is granted ethical approval by the London—Brighton and Sussex Research Ethics Committee (reference: 20/LO/01440). In China, the trial is approved by the medical ethics committees of Capital Medical University, Beijing Tiantan Hospital, Beijing Geriatric Hospital, Chinese People's Liberation Army General Hospital, Taishan Medical University and Xuanwu Hospital. Results will be published in a peer-reviewed journal.

**Trial registration number** ISRCTN15986016.

## Strengths and limitations of this study

► The coach-supported mobile health intervention builds on the previously developed Healthy Ageing Through Internet Counselling in the Elderly electronic health platform, which was shown to improve cardiovascular risk of elderly in the Netherlands, France and Finland.

► The main strengths of the present study are testing the approach in populations with low socioeconomic status or different cultural settings and measuring whether sustained involvement of end users can be achieved, facilitating cultural adaptation of the application.

► Limitations of this study include the impossibility to completely blinding the participants, potentially leading to contamination, and the challenge to sufficiently engage the hard-to-reach target population.

## INTRODUCTION

With global ageing, the prevalence of dementia is expected to increase to over 130 million in 2050, especially in low-income and middle-income countries (LMIC) and in people from low socioeconomic status (SES) background in high-income countries (HIC).[1 2] Strategies need to be developed that aim to reduce the risk of dementia—many of which will be at community and population

BMJ

level, but those that are individually based must be effective, affordable and easily implementable across various healthcare settings.

Up to 40% of dementia cases are estimated to be attributable to potentially modifiable risk factors,[3] of which 10%–20% are cardiovascular risk factors including hypertension, midlife obesity, dyslipidaemia, diabetes, smoking and physical inactivity.[4–6] To date, intervention studies aiming to reduce dementia risk by targeting one or more of these risk factors have shown inconsistent results.[7 8] Results from randomised controlled trials (RCT) on blood pressure-lowering treatment have suggested a beneficial effect on dementia risk, although not consistently and convincingly.[9–11] Since the presence of multiple risk factors may pose an additive or even synergistic effect on dementia risk,[12 13] targeting several risk factors simultaneously may be more effective. The only study to date designed to address this question using dementia as primary outcome did not show a statistically significant effect after 6–8 years of intervention, although subgroup analysis suggested benefit for those with untreated hypertension at baseline.[14]

A considerable challenge when designing a dementia prevention trial is the time lag between the optimal timing of the intervention and the onset of dementia. Using incident dementia as primary outcome requires large sample sizes and/or long follow-up periods to reach statistical power.[15 16] Dementia risk scores could be used as a proxy, especially in trials with follow-up periods up to several years. Another challenge, possibly explaining the neutral results of intervention studies so far, is the small window for risk factor improvement given a background of high-quality cardiovascular risk management in HIC where these studies were performed.[17] This lends further support for targeting people in LMIC and low-SES populations in HIC.

Digital health interventions have the potential to improve cardiovascular risk factors in middle age and beyond, especially when offered with human coaching (blended care).[18] In the Healthy Ageing Through Internet Counselling in the Elderly (HATICE) trial, we recently demonstrated that a coach-supported internet intervention facilitating self-management of cardiovascular risk factors can reduce older adults' cardiovascular risk over a static control platform, both in high and low socioeconomic participant subgroups.[19] Currently, digital health interventions are increasingly offered through smartphones. Smartphone penetration rates are especially high in HIC,[20] also among people with low SES. In 2018, 67% of people with the lowest SES in UK owned a smartphone.[21] Approximately 40%–50% of the LMIC population is connected to mobile internet,[20 22] with rates up to 60% in China.[23] This renders mobile health (mHealth) a promising method for health delivery in underserved populations, including the improvement of cardiovascular risk factors.[24 25]

We have developed a coach-supported mHealth intervention to reduce dementia risk by addressing common cardiovascular risk factors via lifestyle changes, building on the existing HATICE internet platform. Our aim is to assess the effectiveness and implementation of this smartphone intervention on dementia risk in older people at increased risk of dementia from a low-SES population in the UK and from the general population in Beijing, China.

## METHODS

### Study design

Prevention of dementia using mobile phone applications (PRODEMOS) is a multinational, prospective, randomised, open-label blinded endpoint trial with 18-month intervention and follow-up. The study follows a hybrid effectiveness–implementation design, taking a dual focus on assessing effectiveness and implementation outcomes.[26 27] The Amsterdam University Medical Centre (Amsterdam UMC) is the coordinating centre.

### Study population and recruitment

The study population will consist of community-dwelling older adults aged 55–75 years old, of low SES in the UK and of any SES in China, who have ≥2 dementia risk factors and own a smartphone. Low SES in the UK is operationalised as living in a postal code area ranked as equal to or less than the lowest third decile of the index of multiple deprivation.[28] Eligibility criteria are similar for both countries, except for criteria for obesity, based on differences in national prevention guidelines[29] (box 1).

Recruitment will take place in the Eastern Clinical Research Network (National Institutes of Health Research) region of the UK and in the Beijing and Tai'an cities, China. In the UK, recruitment has started in January 2021 and will be undertaken by approximately 10–15 general practioner (GP) practice. A random computer selection of participants living in the designated postal code areas meeting the age criterion and having ≥1 known dementia risk factor according to the GP registry will be approached through an information letter, inviting them to contact the local study centre. In China, participants will be recruited from seven hospitals through advertisements on hospital websites, targeted recruitment via local social media (WeChat) or direct approach by nurses and physicians. In China, recruitment is expected to start mid-2021.

### Intervention and control condition

Central to our study is the PRODEMOS platform, which interconnects the assessor portal, the participant app and the coach portal (figure 1). The assessor portal facilitates blinded collection of baseline and outcome assessments for all participants. The intervention and control condition are both delivered through a smartphone app, which, in the case of intervention participants, allows communication with the coach portal. Data from the assessor portal, participant app and coach portal can be extracted through a researcher portal and stored in

## Box 1 Overview of inclusion and exclusion criteria

**Inclusion criteria**

► Age ≥55 years and ≤75 years old.
► Living in a postal code area ranked as equal to or less than the lowest third decile of IMD*.
► Good proficiency of the national language (English in UK and Mandarin in China).
► Possession of a smartphone.
► ≥ Two dementia risk factors:
  – Insufficient physical activity (self-reported intermediate or vigorous activity of <150 min per week).
  – Active smoking (self-reported use of any sort of tobacco in any quantity).
  – Depression (by meeting at least one of the following criteria):
    – Current diagnosis by specialist or GP.
    – History of treatment for depression (ie, drug therapy or psychotherapy).
  – Manifest cardiovascular disease, as diagnosed by specialist or GP.
  – Diabetes mellitus (by meeting at least one of the following criteria):
    – Diagnosed by specialist or GP.
    – Use of insulin or other blood glucose-lowering medication.
  – Hypertension (by meeting at least one of the following criteria):
    – Diagnosed by specialist or GP.
    – Use of blood pressure-lowering medication.
    – Mean of baseline blood pressure measurements of ≥140 (systolic) or ≥90 (diastolic).
  – Obesity (by meeting at least one of the following criteria):
    – BMI ≥30 (UK) and ≥28 (China).
    – Baseline waist circumference ≥102 cm (men in UK), 90 cm (men in China), 88 cm (women in UK) and 85 cm (women in China).
  – Dyslipidaemia (by meeting at least one of the following criteria):
    – Diagnosed by specialist or GP.
    – Use of lipid-lowering medication.
    – Baseline total cholesterol ≥5.0 mmol/L*.

**Exclusion criteria**

► Manifest dementia, as diagnosed by specialist or GP.
► MMSE <24 (participants with ISCED level of >1) and MMSE <21 (participants with ISCED level of 1).
► Any condition expected to limit 18-month follow-up, including metastasised malignancy or other terminal illnesses.
► Smartphone illiteracy, defined as not being able to send a message from a smartphone.
► Visual impairment interfering with operation of a smartphone.
► Participating in another RCT on behaviour change.
► Present severe alcohol or illicit drug abuse.

*Applies only to participants in the UK.
BMI, Body Mass Index; GP, general practitioner; IMD, Index of Multiple Deprivation; ISCED, International Standard Classification of Education; MMSE, Mini-Mental State Examination; RCT, randomised controlled trial.

a central database. The PRODEMOS platform was built in close collaboration between software developers and researchers from Amsterdam UMC, University of Cambridge, Brighton and Sussex Medical School, Capital Medical University in Beijing, health coaches and representatives of the target population from both countries.

The internet platform previously used in the HATICE trial served as the basis for the PRODEMOS platform.[30] In addition to the transition of the participants' end into a mobile app, adjustments were made to the platform in repeated cycles of interaction with end users. In an iterative process, experiences, needs and wishes from the target population and health coaches regarding the app and coach support, gained through interviews and focus groups, served as a guideline for further development.

Participants have only access to one of two versions of the participant app. Participants randomised to the intervention condition will have access to an interactive smartphone application in their own language (English in the UK and Mandarin in China). The intervention app facilitates coach-supported self-management of seven dementia risk factors, including overweight, unhealthy diet, insufficient physical activity, smoking, hypertension, dyslipidaemia and diabetes. Participants can set personal goals for lifestyle change, following the specific, measurable, achievable, realistic and time-bound principle. Participants receive automated reminders to enter measurements (eg, number of steps and blood pressure) for these goals, facilitating progress monitoring. The intervention participants will receive support from an experienced lifestyle coach, who is trained in motivational interviewing and works according to a coach protocol based on current guidelines for risk factor management. Regular training sessions in each country will enhance uniformity in coaching procedures, taking cultural differences into account. During the baseline visit, after randomisation, the coach discusses the participant's dementia risk profile, and a first lifestyle goal will be set through the app. After the baseline visit, all communication between the participant and coach will take place through the messaging functionality. Through the coach portal, the coach can view goals and measurements, send tailored education modules, and offer remote support to facilitate sustainable behaviour change.

Participants randomised to the control condition will have access to the control app, which is similar in appearance but only contains education material, lacking interactive features and coach-support. During the baseline visit, control participants will receive concise feedback on their risk profile.

The PRODEMOS intervention in its current design is positioned as add-on to existing care.

### Primary and secondary outcomes

Following a type II hybrid design, primary outcomes for effectiveness and implementation are equally important. The primary effectiveness outcome is the change in the Cardiovascular Risk Factors, Ageing and Incidence of Dementia (CAIDE) dementia risk score between baseline and 18-month follow-up.[31] The main secondary effectiveness outcomes include change in the individual modifiable components of the primary outcome, change in ten-year cardiovascular disease (CVD) risk, cost-effectiveness and certain clinical outcomes such as incidence of mild cognitive

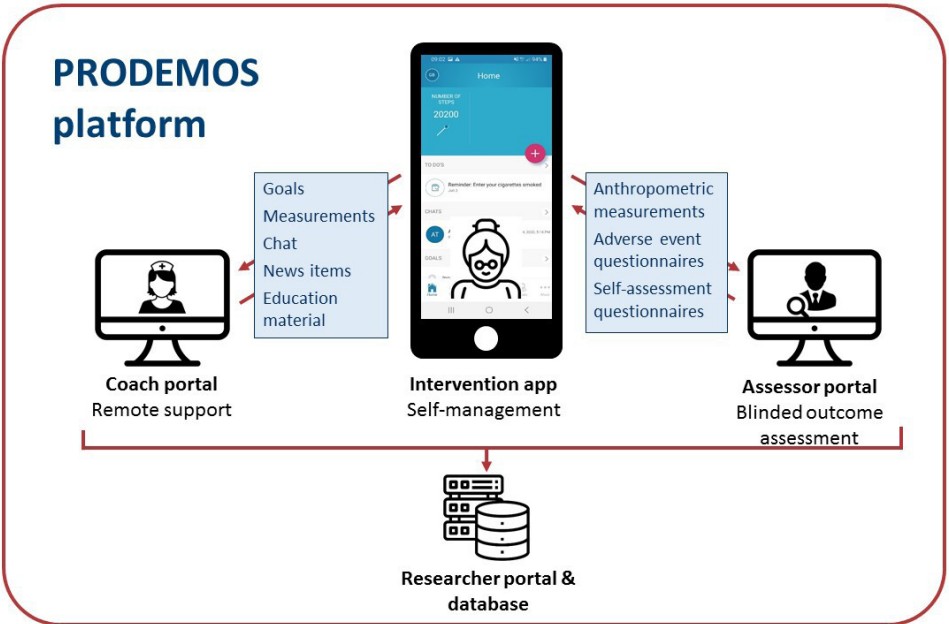

**Figure 1** Overview of the prevention of dementia using mobile phone applications (PRODEMOS) platform and its functionalities. Main features of coach portal: viewing and adjusting details of goals and measurements; sending and receiving chat messages to and from the participants; sending education and news items. Main features of intervention app: setting and adjusting goals; entering measurements; sending and receiving chat messages to and from the coach; reading education and news items automatically pushed by platform or received from the coach; receiving periodic adverse event questionnaires and self-assessment questionnaires. Main features of assessor portal: blinded collection of participant data through electronic case report forms and questionnaires. The control application has similar connections with the assessor portal and the researcher portal/database, but is not connected to the coach portal.

impairment (MCI) and dementia. The operationalisation of all effectiveness outcomes is listed in table 1.

Implementation outcomes include acceptability, adoption, appropriateness, feasibility, fidelity, coverage, sustainability and costs of the implementation. User statistics, including data on goals set, messages sent and education items read, will be analysed to assess adoption and sustained use of the platform. In-depth interviews with participants and coaches will focus on user experiences, particularly with respect to barriers and facilitators for (sustained) platform use. All implementation outcomes and evaluation methods are shown in table 2.

### Study logistics and data collection

The trial design is visualised in figure 2. All participants will receive one phone call and make three visits to a study venue during the study. Data are collected in electronic case report forms that are accessible through the assessor portal (figure 1).

Eligibility criteria that can be assessed remotely will be checked by a local research team member through the screening phone call. During the subsequent screening visit, informed consent (online supplemental file 1) will be obtained, and final eligibility will be assessed by administering the Mini-Mental State Examination; measuring blood pressure, Body Mass Index, waist circumference and total cholesterol (capillary blood sample in the UK; venous blood sample in China); and assessing physical activity, smoking behaviour and a brief medical history. Weight will

be measured with a calibrated scale; blood pressure will be measured twice with a calibrated, automated blood pressure device. Screening visits will be performed by (GP) nurses and local research team members specifically trained to perform these measurements and will take place at the GP surgery or a nearby community venue. Standard operating procedures will be used to achieve uniform measurements within and between countries.

After the screening visit, all participants will fill in eight self-assessment questionnaires in the PRODEMOS app. These questionnaires will be used to assess secondary outcomes (ie, physical activity, quality of life, well-being, disability, depressive symptoms, self-management, anxiety and diet; table 1) and potential barriers for lifestyle behaviour change, which can inform coaches to tailor their coaching strategy. Seven of these questionnaires (ie, International Physical Activity Questionnaire—Short Form, EuroQol Five Dimensions, ICEpop CAPability Measure for Adults, WHO Disability Assessment Schedule 2.0, Geriatric Depression Scale (GDS), Partners In Health and Hospital Anxiety and Depression Scale–Anxiety) have been externally validated in both Western and Chinese populations.[32–45] Owing to obvious cultural differences, we decided to use two different diet questionnaires that were validated in the UK and Chinese population, respectively (Short-Form Food Frequency Questionnaire and Kadoorie Biobank Food Frequency Questionnaire).[46 47]

**Table 1** Effectiveness outcomes

| Primary outcome | Points |
| --- | --- |
| CAIDE Score (range: 0–15), which is composed of and calculated from the following: | |
| Age | |
| <47 years | 0 |
| 47–53 | 3 |
| >53 | 4 |
| Education | |
| ≥10 years | 0 |
| 7–9 years | 2 |
| <7 years | 3 |
| Gender | |
| Female | 0 |
| Male | 1 |
| Systolic blood pressure | |
| ≤140 mm Hg | 0 |
| >140 mm Hg | 2 |
| BMI | |
| ≤30 kg/m$^2$ | 0 |
| >30 kg/m$^2$ | 2 |
| Total cholesterol | |
| ≤6.5 mmol/L | 0 |
| >6.5 mmol/L | 2 |
| Physical activity* | |
| Yes | 0 |
| No | 1 |
| **Secondary outcomes** | |
| Individual modifiable components of the CAIDE score† | Estimated 10-year cardiovascular risk |
| Number of uncontrolled risk factors | LIBRA dementia risk score |
| Active smoking | Number of hospital admissions |
| Medication adherence | Diet‡ |
| Number of drugs | Disability§ |
| Incident dementia¶ | Anxiety** |
| Incident MCI¶ | Self-management†† |
| Incident cardiovascular disease¶,§§ | Depressive symptoms‡‡ |
| Incident diabetes¶ | Quality of life¶¶ |
| All-cause mortality | Cost-effectiveness |

*Assessed according to WHO standard for physical activity of at least 150 min per week.
†Physical activity assessed with the International Physical Activity Questionnaire—Short Form.
‡Assessed with Short-Form Food Frequency Questionnaire (UK) and China Kadoorie Biobank Food Frequency Questionnaire (China).
§Assessed with the WHO Disability Assessment Schedule.
¶Self-reported and cross-checked with general practitioner file.
**Assessed with the Hospital Anxiety and Depression Scale—Anxiety.
††Assessed with the Partners In Health.
‡‡Assessed with the Geriatric Depression Scale.
§§Defined as myocardial infarction or stroke.
¶¶Assessed with the ICEpop CAPability Measure for Adults and EuroQol Five Dimensions Three Levels.
BMI, Body Mass Index; CAIDE, Cardiovascular Risk Factors, Ageing and Incidence of Dementia; LIBRA, Lifestyle for Brain Health; MCI, mild cognitive impairment.

The baseline visit will be conducted face-to-face by the health coach at the GP practice or local community venue. During this visit, self-assessment questionnaires are

reviewed, relevant medical history and medication use are recorded, and participants are randomly assigned to one of the treatment conditions. Only intervention participants will set a first lifestyle goal together with the coach, according to their dementia risk profile.

All participants will receive periodic adverse event (AE) questionnaires in the app, assessing incident dementia, MCI, CVD and diabetes. All self-reported outcomes will be verified with the participant's treating physician.

After 18 months, the questionnaires and all measurements performed during the screening and baseline visit are repeated during the final visit.

### Randomisation and blinding

After completion of the baseline assessments, participants will be individually randomised in a 1:1 ratio, stratified by country, using a central, computer-generated sequence. Participating cohabiting partners will be allocated to the same study condition. Complete blinding of participants is not possible, owing to the nature of the intervention. Participants will be informed that they will be randomised to one of two lifestyle apps, without further details. All outcome assessments will be done by an independent assessor unaware of treatment allocation.

### Safety and privacy

Due to the nature of the intervention, serious AEs are unlikely to occur, and we consider the intervention low risk. A data safety and monitoring board is not installed.

Some precautions are taken to optimise participant safety. First, regardless of their study allocation, participants will be referred to their GP or treating physician if deemed necessary based on their baseline or outcome parameters and local guidelines. Second, AEs will be monitored through three 6 monthly questionnaires, for which participants will receive notifications on their smartphone and reminders through email (UK) or SMS (China). If the participant is not able to fill in the questionnaire, an informant can be contacted. A blinded researcher will, with explicit permission gained through the informed consent procedure, cross-check all reported AEs by consulting the participant's GP or treating physician. Third, the PRODEMOS platform is built in accordance with the highest security requirements in healthcare. It complies with NEN 7510, the Health Insurance Portability and Accountability Act, ISO 133485 and General Data Protection Regulation.

### Protocol adjustments due to COVID-19 pandemic

As a result of the COVID-19 pandemic and related local research restrictions, certain adjustments have been made to the original study protocol as published on the International Standard Randomised Controlled Trial Number Register (ISRCTN). First, recruitment was planned to start in early 2020 but had to be suspended until January 2021. Second, as it is difficult to predict the development of the pandemic and associated restrictions, we have slightly amended the study protocol to allow for flexible

**Table 2** Summary of implementation research methods and outcomes

| Method | Outcome | Measurement | Population* | Timing of assessment |
|---|---|---|---|---|
| Quantitative | Coverage | (Non)Response rates, comparison characteristics of participants with eligible population | Potential target population | At baseline |
| | Adoption | Utilisation, usage, and uptake | Intervention participants, coaches | After 2 weeks |
| | Appropriateness | Short questionnaire of perceived fit or relevance in the target population and the coaches | Intervention participants, coaches | After 3 months and at study end |
| | Acceptability | Short questionnaire of agreeability, user-friendliness, credibility | Intervention participants, coaches | After 3 months and at study end |
| | Sustainability | Adherence, dropout | Intervention participants, dropouts† | Throughout the study |
| | Cost | Implementation costs | N.A. | N.A. |
| Qualitative | Feasibility | The extent to which the mHealth intervention can be carried out →practical and social barriers/facilitators | Intervention participants, dropouts,† coaches | After 3 months and at study end |
| | Appropriateness | Perceived fit or relevance in the target population | Intervention participants, dropouts,† coaches | After 3 months and at study end |
| | Acceptability | Agreeability, user-friendliness, credibility | Intervention participants, dropouts,† coaches | After 3 months and at study end |
| | Fidelity | Degree to which the mHealth application is implemented compared with the original protocol | N.A. | After the study |

*For all analyses, a Chinese and UK population will be involved.
†Study dropouts will be asked to participate in a short exit interview.
mHealth, mobile health.

measurement procedures at baseline that can be operationalised in either one or two face-to-face visits and for a flexible intervention duration of 12–18 months. However, we will strive for a follow-up period of 18 months in as many participants as possible.

### Patient and public involvement

We have received valuable input into the design of the study and mHealth platform from multiple interactive sessions with GPs, health coaches, researchers, representatives of people living with dementia, community leaders and policy makers. Needs and views regarding the intervention were assessed through interviews and focus groups with potential end users in both countries. All patient-facing material used in the UK has been reviewed by potential end users. Qualitative evaluations of the pilot study with research staff, coaches and patient participants were used to refine the intervention and study procedures.

### Statistical analysis
#### Sample size

The CAIDE Score will be used as primary effectiveness outcome. We decided to use a difference of 0.186 points on the CAIDE Score between the average of both study groups as a minimum target threshold, because this difference was observed in the Prevention of Dementia by Intensive Vascular Care trial after 2 years (p=0.005; intervention group=−0.290±1.47 SD and control group=−0.104±1.36 SD). Attrition after 2 years of follow-up was 21% in this study.[14] With 80% power, a 0.05 two-sided significance level, accounting for 21% attrition, and a mean difference in change in CAIDE of 0.186, the required sample size is estimated to be 2319 participants. To allow for unexpected factors, we raised this to 2400.

#### Data analysis

The effect on the CAIDE Score will be analysed using linear mixed-effect models according to the intention-to-treat principle, taking clustering within partner pairs and country into account by testing best fit for random intercept and/or slope. If needed, we will adjust for baseline imbalances and take clustering of the intervention within centre and/or coach into account. No imputation of the CAIDE Score will be done for the primary analysis. In sensitivity analyses, we will use multiple imputation to assess the impact of missing items needed to calculate the CAIDE Score, provided there are no indications that the variables are missing not at random, and a per-protocol analysis for those adherent to the intervention will be performed. Moreover, we will explore the interaction of intervention duration with the effect of the invention by adding an interaction term (intervention duration*randomisation group) to the main model. This will give insight into the potential additional intervention

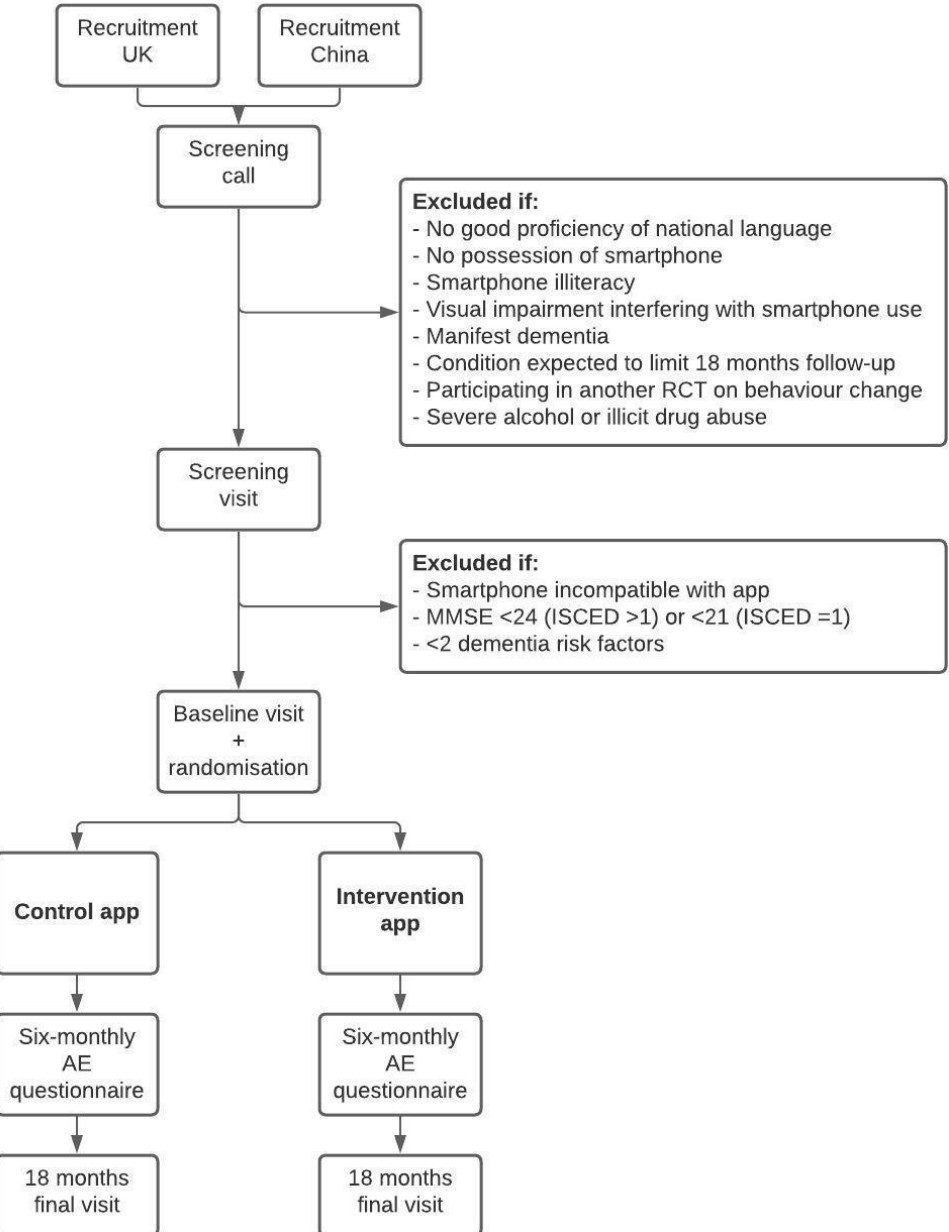

**Figure 2** Trial design. AE, adverse event; ISCED, International Standard Classification of Education; MMSE, Mini-Mental State Examination; RCT, randomised controlled trial.

effect in participants with a follow-up time of less than 18 months.

Subgroup analyses will be performed for country, sex, age group, having a history of CVD, number of risk factors, willingness to change lifestyle (assessed with one question during the baseline visit), participation with(out) a participating partner, having the same coach during the full length of the study and the number of goals set. For all these factors, interaction terms will be included to test for between-subgroup differences in intervention effects.

The effect on individual modifiable components of the CAIDE Score and 10-year CVD Risk Scores will be analysed using linear mixed-effect models according to the intention-to-treat principle, taking clustering within partner pairs and country into account. Self-assessment scales, which are mostly ordinal, will be regarded as linear scales if there are at least four categories and the 'distance' between the categories can be regarded equal. Poisson regression or zero-inflated models may be applied to distributions resembling count or zero-inflated data. The choice of the final model will be a compromise between optimal fit and interpretability of the results for a general clinical public.

Prevalence ratios will be used for self-assessment instruments with defined cut-offs for the presence or absence of a condition, for example, 'depressive symptoms' for a GDS >5. For (clinical) dichotomous outcomes, such as incident CVD, dementia or mortality, Cox proportional hazard models will be used with time using baseline as timescale. A sensitivity analysis will be performed using age as timescale.

The full analysis plan, including the health economic analysis plan entailing the cost–consequence analysis of the within-trial results, the cost-effectiveness analyses and the cost–utility analysis and hypotheses for the subgroup analyses, is published on the ISRCTN website: http://www.isrctn.com/ISRCTN15986016.

## Pilot study

Between December 2019 and March 2020, a 6-week pilot study was conducted in the Brighton and Sussex area, the UK. Since the main aim was to test study logistics and functionality of the intervention app, participants were randomised in a 3:1 (intervention/control) ratio. An invitation letter was sent to 600 potentially eligible patients from two GPs. The response rate was 14.8% (n=89), of whom 21 participants (3.5%) could be included. The main reasons for exclusion were not living in the designated postal code area and having <2 dementia risk factors. Participants had a median age of 69 years old, and 12 (57%) were men. Fifteen participants were allocated to the intervention group and six to the control group.

During the pilot study, 10 of 15 intervention participants set at least one goal (range: 1–8 goals). Goals were set in five domains, including physical activity, healthy diet, body weight, blood pressure and cholesterol. Six of ten participants entered goal-related measurements (range: 2–243 measurements). All intervention participants used the chat functionality to consult the coach. In total, 278 messages were sent back and forth, that is, on average three messages per intervention participant per week.

The pilot study was evaluated through qualitative sessions with the participants and coaches. The main adjustments based on the participants' feedback included improvements to the chat functionality (allowing attachments and larger font size), simplification of the functionality to enter and view measurements, setting the first goal together with the coach and more detailed instructions for app use through an instruction video and written manual. Based on feedback from the coaches, we improved the functionalities for population management in the coach portal, including an input screen to make notes about individual participants and a functionality to send education material to (groups of) participants.

A similar pilot study will be conducted in China, to test platform functionality and study logistics in all seven participating trial centres.

## DISCUSSION

In the PRODEMOS study, we will investigate the implementation of a self-management mHealth intervention with remote coaching and its effect on dementia risk over 18 months. We will target people aged 55–75 years old with elevated dementia risk of low SES in the UK and of any SES in the Beijing and Tai'an cities in China, as these populations are usually not reached by preventive strategies and may benefit the most. User data and qualitative analysis of our pilot study suggest that our mHealth application, after further adaptations to improve attractiveness and usability, is now ready to be studied in older adults who are interested in participating in a study on lifestyle change to lower their overall dementia risk.

The HATICE trial has shown that a coach-supported internet platform can improve cardiovascular risk factors in European elderly. Although we build on these experiences, the modality (ie, app instead of internet platform) and target population are different. The resulting uncertainty that there would be a similar benefit of our intervention renders the use of a hybrid effectiveness–implementation design highly suitable.[27]

## Strengths

Chronic disease risk is largely affected by socioeconomic factors, including psychological, cultural and economic characteristics, requiring preventive strategies that take these aspects into account.[48] In PRODEMOS, we aim to support individuals by offering intensive human support through the app and by aligning the intervention with the healthcare system. In order to eventually embed a complex prevention intervention into primary healthcare, it is crucial to involve and consult all stakeholders, such as GPs, practice nurses, and end users.[49] In the current hybrid effectiveness–implementation study, we take some first steps to explore the possibilities and challenges for embedding the intervention in existing healthcare. This study will provide concrete evidence of the scale of the change that might be achieved for individuals at risk, whether and how this approach is taken up within diverse populations.

The PRODEMOS study is designed as one trial, recruiting participants in two different countries, increasing the external validity of the results. Overall, both countries will follow the same research protocol and highly similar standard operating procedures and will investigate similar interventions. Through semistructured interviews among the elderly in Beijing and the UK, we learnt that needs and wishes regarding lifestyle behaviour change through mHealth are largely similar (manuscripts currently being drafted). Therefore, the Chinese and UK intervention will share the same functionalities and coaching procedures. Given obvious cultural-related and healthcare-related differences, certain aspects of the study logistics, lifestyle support and layout of the app had to be culturally adjusted. In a preplanned subgroup analysis, we will assess both effectiveness and implementation outcomes for both countries separately.

## Limitations

The study may yield some limitations. First, the optimal age range for trials on dementia risk reduction is unknown.[15] There is a trade-off between potentially more effective treatments in midlife and the chance to detect treatment effects on cognitive outcomes in late life.[4] As in the current study, we are assessing both a dementia

risk score and clinical outcomes; we have taken a pragmatic approach, targeting individuals aged 55–75 years old[15].

Second, change in CAIDE dementia risk score is not easily translated into incidence of dementia. However, although not specifically designed as RCT outcome measure, the CAIDE Score can detect change over time.[50]

A third potential limitation is that, owing to the nature of the intervention, blinding of the participants is only partly possible. A certain degree of contamination might occur, especially in communities that live closely together. The study logistics and intervention are designed in such a way as to limit contact between participants after randomisation.

Finally, the results of the baseline measurements will be revealed to all participants, potentially leading to treatment effects in both study conditions. Also, behaviour of participants and their treating physicians may change in both study conditions as a reaction to the awareness of being part of the study (Hawthorne effect). Both mechanisms will perhaps mask (part of) the 'true' contrast in dementia risk between the intervention and control condition.

For the planned health economic analyses, we will rely on economic modelling, based on the intermediate outcomes reflecting risk of dementia and CVD and assumptions on their causality with the clinical endpoints dementia and CVD, because the study is not designed nor powered to detect an effect on these clinical endpoints.

The high prevalence of dementia, lower provision of high-quality cardiovascular preventive care in LMIC and lower uptake of such programmes in Western low-SES populations require affordable and straightforward preventive strategies. If proven effective and implementable, our pragmatic smartphone intervention facilitates widespread use and reduction of dementia risk for hard-to-reach populations across the globe.

## ETHICS AND DISSEMINATION

The PRODEMOS trial is sponsored in the UK by the University of Cambridge and is granted ethical approval by the London–Brighton and Sussex Research Ethics Committee (reference: 20/LO/01440). In China, the trial is approved by the medical ethics committees of Capital Medical University, Beijing Tiantan Hospital, Beijing Geriatric Hospital, Chinese People's Liberation Army General Hospital, Taishan Medical University and Xuanwu Hospital. Data will be exported in a pseudonymised format according to prevailing guidelines on good clinical practice in both countries. Only anonymised data will be exchanged between the UK, China and the Netherlands. The exported data will be stored centrally on a protected server in the Netherlands, which is compatible with the highest standards of data management in medical research. Results will be published in a peer-reviewed journal.

**Author affiliations**
[1]Department of General Practice, Amsterdam UMC Locatie AMC, Amsterdam, The Netherlands
[2]Department of Neurology, Amsterdam UMC Locatie AMC, Amsterdam, The Netherlands
[3]Beijing Key Laboratory of Clinical Epidemiology, Capital Medical University School of Public Health, Beijing, China
[4]Edith Cowan University School of Medical and Health Sciences, Joondalup, Western Australia, Australia
[5]INSERM-University of Toulouse UMR1027, Toulouse, France
[6]Department of Epidemiology and Public Health, Toulouse University Hospital, Toulouse, France
[7]Cambridge Public Health, University of Cambridge, Cambridge, UK
[8]Alzheimer Europe, Luxembourg
[9]Department of Primary Care and Public Health, Brighton and Sussex Medical School, Brighton, UK
[10]Vital Health Software, Ede, The Netherlands
[11]Department of Public and Occupational Health, Amsterdam UMC Locatie AMC, Amsterdam, The Netherlands
[12]Alzheimer Centre Limburg, School for Mental Health and Neuroscience, Maastricht University Medical Centre, Maastricht, The Netherlands
[13]Division of Neurogeriatrics, Department of Neurobiology, Care Sciences and Society, Karolinska Institutet, Stockholm, Sweden
[14]School of Public Health, Shandong First Medical University and Shandong Academy of Medical Science, Tai'an, China
[15]Beijing Neurosurgical Institute, Beijing, China
[16]Center for Cognitive Disorders, Beijing Geriatric Hospital, Beijing, China
[17]Department of Geriatrics, The Second Medical Centre and National Clinical Research Centre for Geriatric Diseases, Chinese PLA General Hospital, Beijing, China
[18]Fuzhou Comvee Network & Technology Co., Ltd, Fuzhou, China
[19]Department of Neurology, Xuanwu Hospital, Capital Medical University, Beijing, China
[20]Health Management Institute, The Second Medical Centre and National Clinical Research Centre for Geriatric Diseases, Chinese PLA General Hospital, Beijing, China
[21]Centre for Cognitive Neurology, Department of Neurology, Beijing Tiantan Hospital, Capital Medical University, Beijing, China
[22]Department of Neurology, Radboud University Donders Institute for Brain, Cognition and Behaviour, Nijmegen, The Netherlands

**Acknowledgements** The authors thank the European Union's Horizon 2020 Research and Innovation Programme and National Key R&D Programme of China for the funding of the trial. The authors thank Lonneke van Vught for her substantial contribution in the early phase of the trial design. The authors thank Kevin Hekert, Louwrens Knulst and Michiel van Dam for their substantial contribution in building the platform. The authors thank Jinxia Zhang, Xiaoyu Zhang, Mingyang Cao, Bin Jiang, Siqi Ge, Maolong Gao, Mo Li, Nayan Huang, Danning Li, Mingyue He, Weijiao Zhang, Huiying Guan, Jinghui Li, Yan Gong, Na Niu, Xiang Jia, Fei Wang, Jing Sun, Liyong Wu, Dan Li, Baoliang Sun, Hui Yuan, Guohua Wang, Xizhu Xu, Cancan Li, Wenran Zhang, Juan Du and Libin Song for their contributions to the trial logistics. The authors thank all patient participants and other stakeholders who took part in patient and public involvement groups for their contribution to the platform and study design.

**Contributors** EE was responsible for the drafting of the manuscript. ER, EPMvC, WAvG, MPH-B, CBr and WeiW were responsible for the study conception. ER, EPMvC, MPH-B, MS, SA, CBi, NC, EF, JG, WAvG, HvM, WenW, YW, AW, WeiW and CBr were responsible for the design of the trial. EE, MH, LEB, RLB, AvdG, RH, HH, DL, HL, JL, MvdM, YN, SS, XY, YY, QZ and WZ were involved in trial design and coordination. All other authors were responsible for critically revising the manuscript. All authors approved the final version of the manuscript. WeiW, CBr, EPMvC and ER are shared last authors.

**Funding** This project has received funding from the European Union's Horizon 2020 Research and Innovation Programme under grant agreement No. 779 238 and the National Key R&D Programme of China (2017YFE0118800).

**Competing interests** None declared.

**Patient consent for publication** Not required.

**Provenance and peer review** Not commissioned; externally peer reviewed.

**ORCID iDs**
Esmé Eggink http://orcid.org/0000-0001-7132-2937
Nicola Coley http://orcid.org/0000-0002-1671-824X
Elizabeth Ford http://orcid.org/0000-0001-5613-8509
Jihui Lyu http://orcid.org/0000-0003-1035-6943
Harm van Marwijk http://orcid.org/0000-0001-6206-485X
Wenzhi Wang http://orcid.org/0000-0002-0086-1121

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
