## [Reviewer comments · BMJ Open]

ARTICLE DETAILS

TITLE (PROVISIONAL)	Prevention of Dementia using Mobile Phone Applications (PRODEMOS): Protocol for an international randomised controlled trial
AUTHORS	Eggink, Esmé; Hafdi, Melanie; Hoevenaar-Blom, Marieke; Song, Manshu; Andrieu, Sandrine; Barnes, Linda E; Birck, Cindy; Brooks, Rachael L; Coley, Nicola; Ford, Elizabeth; Georges, Jean; van der Groep, Abraham; van Gool, Willem A; Handels, Ron; HOU, Haifeng; Li, Dong; Liu, Hongmei; Lyu, Jihui; van Marwijk, Harm; van der Meijden, Mark; Niu, Yixuan; Sadhwani, Shanu; Wang, Wenzhi; Wang, Youxin; Wimo, Anders; Ye, Xiaoyan; Yu, Yueyi; Zeng, Qiang; Zhang, Wei; Wang, Wei; Brayne, Carol; Moll van Charante, Eric P; Richard, Edo

VERSION 1 – REVIEW

REVIEWER	Yousaf, Kanwal University of Engineering and Technology
REVIEW RETURNED	22-Feb-2021

GENERAL COMMENTS	In this protocol study, the authors will investigate the effectiveness of the coach-supported mHealth dementia app to reduce dementia risks by addressing cardiovascular risk factors. The study is important and up to date. The paper is well-written, and some constructive suggestions for the paper are:  1. In Fig. 02, the exclusion criteria, after screening call and screening visit, also be illustrated in the diagram. 2. What are the main features of PRODEMOS app and what platform-supported? 3. A very comprehensive methodology for the study is defined. But it needs some additions: Page 11, line 34: What are the 8-self assessment questionnaires, that will be filled by the participants, as these questionnaires will be used for further assessment? The authors should provide the list of questionnaires used in this study. Page 12, line 59: it states that "... was planned to start in early 2020, but is now foreseen to start in January 2021...". This statement needs changes. If the study is started then mention the started date. 4. Lastly, add more details in the conclusion section.
--

REVIEWER	Faieta, Julie The Ohio State University Wexner Medical Center
REVIEW RETURNED	03-Mar-2021

GENERAL COMMENTS	This protocol describes a study that is highly important and timely! The use of pervasive technologies within healthcare continues to grow, and there is increasing potential for mHealth mediated health support for those living with or at risk of developing dementia. A few specific thoughts are outlined below:  - Great overview of the potential impact and timelines of mobile app based interventions; I appreciate the statistics relative to the UK populations (page 7, lines 57-59). However, I think adding additional statistics for other people groups would add to the perceived value of this project, especially with the discussion of LIMC in the earlier in the introduction section. - The risk factors addressed with this app are very important areas to consider (page 10, lines 14-17), I do wonder why sleep health was not included, particularly since that is an area that has been targeted by many commercially available apps. - If there will be any ability to objectively measure level of engagement with the mHealth I think this would strengthen the methods of the study (perhaps something along the lines of time of interaction/week or number of times looking in per day). - In the Patient and Public involvement section, I would like to know a bit more about the profiles of “stakeholders” who where involved in the refinement of the mHealth interface- were they individuals with dementia, at risk of dementia, family members, app developers, clinicians? I think incorporating these stakeholders is a very important step in the human centered design process of the project app, I would just like to better understand the specifics of this step.
---

VERSION 1 – AUTHOR RESPONSE

Reviewer 1

Comment 1

In Fig. 02, the exclusion criteria, after screening call and screening visit, also be illustrated in the diagram.

Response

We thank the reviewer for this comment. We agree that figure 2 can benefit from additional information regarding the exclusion criteria and have expanded the flowchart accordingly. We have not included age and postal code region as eligibility criteria in this figure, as those criteria are already included in the initial computer selection (Methods – Study population and recruitment - rules 157-160).

Changes

- Results section, figure 2.

Comment 2

What are the main features of PRODEMOS app and what platform-supported?

Response

We thank the reviewer for this comment. The participant app, coach portal, and assessor portal together form the PRODEMOS platform (Methods – Intervention and control condition - rules 166-167). As we have tried to capture all components of the PRODEMOS platform in figure 1, we understand and agree that the specific features of each component may need further clarification. In response to the reviewer’s comment, we have expanded the legend of figure 1 with the main features of each platform component.

Moreover, we have made a textual adjustment to subheading “Intervention and control condition” with the aim to clarify that participants have only access to the app (i.e. not to other parts of the platform).

Changes

- Methods section, subheading “Intervention and control condition”, rule 181.
- Results section, legend of figure 1.

Comment 3

A very comprehensive methodology for the study is defined. But it needs some additions:

Page 11, line 34: What are the 8-self assessment questionnaires that will be filled by the participants, as these questionnaires will be used for further assessment? The authors should provide the list of questionnaires used in this study.

Page 12, line 59: it states that "... was planned to start in early 2020, but is now foreseen to start in January 2021...". This statement needs changes. If the study is started then mention the start date.

Response

We fully agree with these two proposed additions to the methodology section. We have specified the secondary outcomes and associated self-assessment questionnaires in the text under subheading “Study logistics and data collection”. Moreover, we have adjusted the details on the start of recruitment under subheading “Study population and recruitment” and under subheading “Protocol adjustments due to COVID-19 pandemic”.

Changes

- Methods section, subheading “Study logistics and data collection”, rules 233-234, 236-239, and 242-243.
- Methods section, subheading “Study population and recruitment”, rules 156-157, 162, and 163.
- Methods section, subheading “Protocol adjustments due to COVID-19 pandemic”, rule 281.

Comment 4

Lastly, add more details in the conclusion section.

Response

We thank the reviewer for this request to elaborate in the conclusion section. However, the editor has asked us to remove the conclusion section, as this is not required/suited for this type of manuscript. We have therefore removed the conclusion paragraph and have merged this with the discussion section.

Changes

- Removed conclusion as a separate section, and merged with discussion section (rules 434-438).

Reviewer 2

Comment 1

Great overview of the potential impact and timelines of mobile app based interventions; I appreciate the statistics relative to the UK populations (page 7, lines 57-59). However, I think adding additional statistics for other people groups would add to the perceived value of this project, especially with the discussion of LIMC in the earlier in the introduction section.

Response

We thank the reviewer for the kind words and the useful feedback. We have slightly expanded the overview of statistics regarding mobile internet connection around the world, with a special focus on individuals in LMIC (China in specific) and (low SES individuals in) the UK.

Changes

- Introduction section, second last paragraph, rules 130-132.

Comment 2

The risk factors addressed with this app are very important areas to consider (page 10, lines 14-17), I do wonder why sleep health was not included, particularly since that is an area that has been targeted by many commercially available apps.

Response

We agree with the reviewer that many commercially available apps (also) target other lifestyle-related factors, such as sleep quality. For this effectiveness-implementation study, we selected risk factors based on two criteria: 1) risk factors should be measurable & modifiable by lifestyle changes within 18 months of intervention, and 2) there should be a solid evidence base for the relationship between the risk factors and dementia risk. For the latter criterion, we adhere to high-quality reviews and meta-analyses in the field, including the review by Barnes et al. 2011 (The projected effect of risk factor reduction on Alzheimer's disease prevalence) and the Lancet report from Livingston et al. 2020 (Dementia prevention, intervention, and care: 2020 report of the Lancet commission). Based on results from these studies, we did not select sleep quality as a risk factor in our intervention.

Comment 3

If there will be any ability to objectively measure level of engagement with the mHealth I think this would strengthen the methods of the study (perhaps something along the lines of time of interaction/week or number of times looking in per day).

Response

To objectively assess the implementation outcomes, we will indeed collect and extract data on user statistics. It is technically difficult to reliably record app log-ins / actual time of use. Therefore, we will collect specific data on number of goals set, number of chat messages sent, and number of education items read as proxies for actual app use and engagement. In the methods section, subheading "Primary and secondary outcomes", we have added these details on the user statistics.

Changes

- Methods section, "Primary and secondary outcomes", rules 212-214.

Comment 4

In the Patient and Public involvement section, I would like to know a bit more about the profiles of "stakeholders" who were involved in the refinement of the mHealth interface- were they individuals with dementia, at risk of dementia, family members, app developers, clinicians? I think incorporating these stakeholders is a very important step in the human centered design process of the project app, I would just like to better understand the specifics of this step.

Response

We thank the reviewer for this comment. We have provided more detail regarding all stakeholders involved in the development and refinement of the intervention in the methods section, subheading "Patient and Public Involvement".

Changes

- Methods section, subheading "Patient and Public Involvement", rules 290-291.